# Effect of Fly Maggot Protein as Dietary on Growth and Intestinal Microbial Community of Pacific White Shrimp *Litopenaeus vannamei*

**DOI:** 10.3390/biology12111433

**Published:** 2023-11-15

**Authors:** Xintao Li, Lishi Yang, Shigui Jiang, Falin Zhou, Song Jiang, Yundong Li, Xu Chen, Qibin Yang, Yafei Duan, Jianhua Huang

**Affiliations:** 1Key Laboratory of South China Sea Fishery Resources Exploitation and Utilization, Ministry of Agriculture and Rural Affairs, South China Sea Fisheries Research Institute, Chinese Academy of Fishery Sciences, Guangzhou 510300, China; lixt597@163.com (X.L.); yangls2016@163.com (L.Y.); jiangsg@21cn.com (S.J.); zhoufalin0925@163.com (F.Z.); tojiangsong@163.com (S.J.); liyd2019@163.com (Y.L.); duanyafei89@163.com (Y.D.); 2College of Fisheries and Life Science, Shanghai Ocean University, Shanghai 201306, China; 3Shenzhen Base of South China Sea Fisheries Research Institute, Chinese Academy of Fishery Sciences, Shenzhen 518108, China; 4Tropical Fishery Research and Development Center, South China Sea Fisheries Research Institute, Chinese Academy of Fishery Sciences, Sanya 572018, China; chenxu7495@163.com (X.C.); yangqibin1208@163.com (Q.Y.)

**Keywords:** *Litopenaeus vannamei*, fly maggot protein, intestinal microorganisms, growth performance, survival rate

## Abstract

**Simple Summary:**

Fly maggots are nutrient-rich, especially with regard to their protein content, making them an ideal replacement for the more expensive fishmeal. We aimed to analyse the effects of different treatments (i.e., fresh fly and fermented fly maggot protein feed, and fermented fly maggot protein feed with high-temperature pelleting) of fly maggot protein feed on the growth and intestinal flora of Pacific white shrimp (*Litopenaeus vannamei*). Fresh fly maggot protein in the feed was detrimental to shrimp growth, whereas fermented and high-temperature pelleted fly maggot protein feeds improved shrimp growth and survival. In addition, the fermented and high-temperature pelleted fly maggot protein feed increased the abundance of beneficial bacteria in shrimp intestines and reduced the growth of harmful bacteria. In contrast, fresh fly maggot proteins led to invasion by *Vibrio* as well as antibiotic-resistant bacteria. This study contributes to our understanding of the regulatory effects of different treatments on the growth and intestinal microflora of *L. vannamei* and provides a basis for the use of fly maggots as a source of protein in the context of aquaculture.

**Abstract:**

As the intensive development of aquaculture persists, the demand for fishmeal continues to grow; however, since fishery resources are limited, the price of fishmeal remains high. Therefore, there is an urgent need to develop new sources of protein. They are rich in proteins, fatty acids, amino acids, chitin, vitamins, minerals, and antibacterial substances. Maggot meal-based diet is an ideal source of high-quality animal protein and a new type of protein-based immune enhancer with good application prospects in animal husbandry and aquaculture. In the present study, we investigated the effects of three different diets containing maggot protein on the growth and intestinal microflora of *Litopenaeus vannamei*. The shrimp were fed either a control feed (no fly maggot protein added), FM feed (compound feed with 30% fresh fly maggot protein added), FF feed (fermented fly maggot protein), or HT feed (high-temperature pelleted fly maggot protein) for eight weeks. The results showed that fresh fly maggot protein in the feed was detrimental to shrimp growth, whereas fermented and high-temperature-pelleted fly maggot protein improved shrimp growth and survival. The effects of different fly maggot protein treatments on the intestinal microbiota of *L. vannamei* also varied. Fermented fly maggot protein feed and high-temperature-pelleted fly maggot protein feed increased the relative abundance of *Ruegeria* and *Pseudomonas*, which increased the abundance of beneficial bacteria and thus inhibited the growth of harmful bacteria. In contrast, fresh fly maggot proteins alter the intestinal microbiome, disrupting symbiotic relationships between bacteria, and causing invasion by *Vibrio* and antibiotic-resistant bacteria. These results suggest that fresh fly maggot proteins affect the composition of intestinal microorganisms, which is detrimental to the intestinal tract of *L. vannamei*, whereas fermented fly maggot protein feed affected the growth of *L. vannamei* positively by improving the composition of intestinal microorganisms.

## 1. Introduction

The Pacific white shrimp *Litopenaeus vannamei* is a globally important aquatic species and one of the most productive shrimp species, providing a high-quality protein source for the human diet. Shrimp are omnivorous and carnivorous animals, and proteins in the compound feed are the primary nutrients that determine shrimp growth, development, and performance [1]. To meet the protein demand of aquatic animals, protein levels in general commercial formulations usually account for 25–50%. Accompanied by a continuous expansion of their feeding scale, the demand for fishmeal in the feed has increased continuously. Due to the decline in fish meal production, rising prices, and unstable supply caused by factors such as declining wild resources and climate change, the search for alternative protein sources of good quality and abundance to replace fish meal has become a popular research topic in recent years. However, with the increasing scale and intensification of farming, shrimp diseases have become more frequent, causing serious economic losses to the shrimp farming industry. Although chemical drugs and antibiotics affect disease prevention and treatment, they may lead to negative effects, such as environmental pollution, drug residues, generation of resistant strains, food safety risks, and risks to human health. Therefore, functional feed research aims to replace fish meal proteins and improve the nutrition, immunity, and organismal defence of shrimp, which are the key areas of research for aquafeed companies.

The larvae of the house fly (*Musca domestica*) and black gadfly (*Hermetia illucens* L.) (commonly known as fly maggots) have the advantages of large numbers, wide distribution, strong reproductive capacity, and low rearing costs. The crude protein content of the dry matter obtained after processing the larvae is close to fish meal (as high as 58.80%–63.89%) [2], which has approximately the same protein content as fish meal [3], and fly maggots are rich in protein, fatty acids, amino acids, chitin, vitamins, minerals, and antimicrobial active substances [2]. Maggot meal-based diets not only serve as a desirable animal protein feed source but also as a new type of protein-based immune enhancer with good application prospects in livestock and aquaculture [4,5]. Recent studies have shown that fly maggots can improve the growth performance of *L. vannamei*, including body composition [6] fatness, and survival [7], and increase immune enzyme activity [8] and immunity [9]. Furthermore, fly maggot protein improves appetite, weight gain, survival rate, and immune function in *Trionyx sinensis* [10] and tilapia (*Oreochromis niloticus*) [11]. Several studies [12] have shown that partially replacing fish meal with fly maggot meal as artificial bait improves the total feed consumption, feed conversion efficiency, protein efficiency ratio, specific growth rate, and survival rate of common carp (*Cyprinus carpio* L.). In contrast, the use of fly maggot protein as a substitute for fish meal in the rearing of clariid catfish (*Clarias gariepinus*) had no substantial effect on growth performance [13]. These findings suggest that fly maggots can partially replace fish meal in commercial feed to improve the growth performance and immunity of aquaculture animals. However, the effects on these functions are closely related to the manner in which the feed is used. Although the direct ingestion of fresh fly maggots is better for nutritional utilisation, some fly maggots float on the water surface and are not easily ingested, which makes them easy to pollute aquaculture water in the long term. Fly maggot powder is crushed after high-temperature drying, which destroys its active substances and immunomodulatory functions. Fly maggot antibacterial peptides are extremely expensive after being processed using extraction, separation, and drying processes [6]. Therefore, a less costly method for treating fly maggot proteins is required to fully utilise their benefits.

With the ban on antibiotics, research on alternative products has been receiving attention in the livestock, poultry, and aquaculture industries, and microbially fermented feed is one such product. Many studies have shown that the use of fermented feed can improve the content of feed proteins and vitamins [14], reduce mycotoxins and other toxic and harmful substances, and improve the safety and effectiveness of the feed and the growth performance [15]. Furthermore, it promotes the digestion and absorption of nutrients, improves the conversion rate of feed nutrients [14], balances intestinal bacteria, promotes intestinal peristalsis, and prevents diarrhoea and constipation [16]. Fermented feeds improve the diversity index of intestinal microflora and the abundance of Lactobacillus in shrimp, thereby improving the composition of the intestinal microflora of *L. vannamei* [17]. 

Several factors, including the food type and intestinal microbiota, influence shrimp growth. The intestine is the most important digestive and absorptive organ of shrimp, and diet strongly alters the composition of the intestinal microbiota, which may result in the production of microbiota metabolites [18]. The intestine is inhabited by several microorganisms that influence host digestion and immunity. Stable microbial communities can promote host health by producing beneficial metabolites and immune stimulation [19]. Therefore, intestinal microbial changes and growth performance can be used to assess the effects of fly maggot protein and fermented feed. Based on previous studies, we hypothesised that different treatments with fly maggot protein might affect the growth, survival, and intestinal microbial balance of *L. vannamei*. In this study, we compared the effects of regular compound feed, regular compound feed supplemented with 30% fresh fly maggot protein, fly maggot protein-fermented feed, and high-temperature-pelleted fly maggot protein feed on the growth of *L. vannamei* and evaluated the effects of different treatment methods on the application of fly maggot protein in shrimp feed using intestinal microflora analysis. Our results provide a basis for the application of fly maggot protein in the context of *L. vannamei* production.

## 2. Methods and Materials

### 2.1. Shrimp and Culture Conditions 

*L. vannamei* were collected from a semi-intensive culture pond at the Shenzhen Base, South China Sea Fisheries Research Institute of the Chinese Academy of Fishery Sciences (Shenzhen, China). Before the experiment, the shrimp were acclimated to the experimental tanks for seven days. A total of 600 shrimp of uniform size, with an average weight of 3.970 ± 0.074 g, were selected for the experiment. The shrimp were randomly allocated to 12 glass-fibre-reinforced plastic water tanks with a capacity of 500 L (1.3 m × 1.0 × 0.4 m), each containing 300 L of water and 60 shrimp. The shrimp were cultured in filtered and aerated seawater (salinity 30‰, pH 8.1–8.2, temperature 30 ± 0.5 °C, and dissolved oxygen > 6.0 mg/L) with concentrations of ammonia-N < 0.1 mg/L and nitrite-N < 0.01 mg/L. Shrimp were fed commercial feed three times daily at 5% body weight.

### 2.2. Diet Preparation

The basic feed for this experiment was a common compound feed, and three experimental feeds were prepared: fresh fly maggots were cleaned, decontaminated, sterilised, cleaned using an ozone cell, and milled to obtain the fly maggot protein source (BoYuJia Biotechnology, Shenzhen, China). No fish meal was added to any of the experimental feeds, and the normal feed without fish meal was mixed with 30% fresh fly maggot protein, fly maggot protein-fermented feed (feed ingredients with 30% fresh fly maggot protein fermentation treatment, no high-temperature pelleting), or fly maggot protein pellet feed (feed ingredients with 30% fresh fly maggot protein fermentation feed, high-temperature pelleting). The feed formulation (Appendix A), amino acid composition (Appendix A), and nutritional composition (Appendix A) are available in the supplementary materials. Methionine and cystine were hydrolysed using oxidative acid hydrolysis, and the rest of the amino acids were hydrolysed via acid hydrolysis. Each treatment was tested in triplicate to obtain the average value, and the routine nutrient composition of the feeds was tested using the AOAC method [20]. Among them, the crude protein content was determined using a Kjeltec^TM^2300 Kjeldahl nitrogen meter (FOSS, Denmark); crude fibre content was determined from the weight of dried residue after extracting the feed with acid and alkali; calcium ion content was determined using disodium ethylenediaminetetraacetic acid complexometric titration; phosphorus content was determined using spectrophotometry; and crude ash content was determined using scorching in a Marfo oven at 550 °C. Moisture content was determined using the constant temperature drying method at 105 °C. All experimental feeds were stored below −20 °C before use. The feed was removed from the −20 °C refrigerator and left at room temperature for half an hour before feeding. 

### 2.3. Experimental Design and Sample Collection

After seven days of acclimatisation, the shrimp were randomly divided into four groups: CK, FM, FF, and HT. Three replicate tanks of 50 shrimp each were used. The CK group was fed a common compound feed, the FM group was fed a diet containing 30% fresh fly maggot protein, the FF group was fed a fermented diet containing 30% fly maggot protein, and the HT group was fed a high-temperature pelletised fermented diet containing 30% fly maggot protein. The feeding trial lasted eight weeks. The four diets were fed at 3% shrimp body weight three times a day (08:00, 12:00, and 17:00). Shrimp intake was near saturation during the feeding trials. The shrimp were fed according to their appetite, and uneaten feed and faeces were cleared from the tanks 1 h after feeding. 

After the eight-week feeding trial, the growth and survival of shrimp in each tank were assessed and randomly sampled. Considering inter-individual variations, five intestines from each tank were combined into one intestinal microbial sample and one gut microbial sample. All the samples were flash-frozen in liquid nitrogen prior to analysis.

### 2.4. Growth Performance Analysis

At the end of the experiment, the shrimp were weighed to determine the weight gain rate (WGR), specific growth rate (SGR), and survival rate (SR). The growth performance and survival of *L. vannamei* in all the groups were calculated using the following equations [21]:Weight gain rate (%) = 100 × (final weight − initial weight)/initial weight. 
Specific growth rate (%/d) = 100 × (ln end weight − ln initial weight)/number of days of feeding
Survival rate (%) = 100 × (initial number of prawns − number of dead prawns)/initial number of prawns 

### 2.5. DNA Extraction, PCR Amplification, and Sequencing

Microbial DNA from intestinal samples was extracted using the Stool Genomic DNA Extraction Kit (DP712) (TIANGEN, Beijing, China)and the magnetic bead soil method, according to the manufacturer’s protocols. A NanoDrop 2000 UV-vis spectrophotometre (Thermo Scientific, Wilmington, DE, USA) was used to determine the DNA concentration and purification. The amplification of the hypervariable V3-V4 region of the bacterial 16S rRNA gene was conducted using a thermal cycling PCR system (GeneAmp 9700, ABI, Bio-Rad, Hercules, CA, USA) and the 341F (5′-CCTAYGGGRBGCASCAG-3′) and 806R (5′-GGACTACNNGGGGTATCTAAT-3′) primers. The PCR cycling parameters were as follows: pre-denaturation at 95 °C for 3 min; 30 cycles of denaturation at 95 °C for 30 s, annealing at 55 °C for 30 s, an extension at 72 °C for 45 s, and a final extension at 72 °C for 10 min (PCR instrument: ABI GeneAmp 9700, Bio-Rad, USA). The PCR reactions (20 μL mixture containing 4 μL of 5 × FastPfu Buffer, 2 μL of 2.5 mM dNTPs, 0.8 μL of each primer (5 μM), 0.4 μL of FastPfu Polymerase, and 10 ng of template DNA) were performed in triplicate. The PCR products were recovered using a 2% agarose gel, purified using the AxyPrep DNA Gel Extraction Kit (Axygen Biosciences, Union City, CA, USA), eluted in Tris-HCl, and detected by 2% agarose electrophoresis. The detection and quantification were conducted using QuantiFluor ~ST (Promega, Madison, WI, USA). Purified amplified segments were used for library construction following standard operating procedures for the Illumina MiSeq platform (Illumina, San Diego, CA, USA). Sequencing was performed using the Illumina MiSeq PE300 platform.

### 2.6. Bioinformatics Analysis

First, QIIME2 was used for the quality control of mouse microbiome sequencing data to detect variability in amplicon sequences [22]. Next, we obtained classification tables for species annotation by comparing the current ASV sequences with those in the Greengenes (16 S rRNA) database [23]. Each sequence in the Silva 132 database was analysed for classification using the RDP classifier algorithm, and only results with a confidence level above 70% were considered. Based on the results of a recent study, the mi-biome diversity of four groups of experimental animals was analysed by evaluating α-diversity indices (including Chao1, Simpson, and Shannon indices) as well as β-diversity metrics (using the principal coordinate analysis method) [24]. Linear discriminant analysis (LDA) and effect size (LEfSe) analyses were performed to identify factors with notable effects by identifying bacterial taxa that differed between the groups. In this analysis, the LDA score threshold of 3.0. was considered significant [25]. In addition, network analysis was used to reveal the differences in the microbiota among the four shrimp groups [4,26,27]. Ultimately, using the PICRUSt and MetaCyc databases for annotation, we predicted the potential function of KEGG homologues (KO) that may be present in shrimp microbiota [28]. Unless otherwise indicated, the parameters used in these analyses were set to default [23].

### 2.7. Correlations between Intestine Bacteria and the Growth of Shrimp

Redundancy analysis (RDA) was performed using the R package ‘vegan’ to reveal potential associations between microbial communities and related environmental factors. Co-occurrence analysis was performed by calculating Spearman’s rank correlations among the taxa and network diagrams were used to show the associations among the taxa.

### 2.8. Statistical Analysis

The data were expressed as the mean ± SD of each variable. Statistical significance was determined using a one-way analysis of variance (ANOVA) and Duncan’s multiple range test (SPSS version 22.0). Statistical significance was set at *p* < 0.05. significant.

## 3. Results

### 3.1. The Growth and Survival of the Shrimp

Compared to the CK group, there were significant differences in WGR, SGR, and SR among the test groups (Table 1). In terms of weight gain rate, the HT group had a significantly higher WGR than the FM and FF groups (*p* < 0.05) but lower than that of the CK group (*p* > 0.05). The HT group had the highest SR (*p* < 0.05).

### 3.2. Intestine Microbial Richness and Diversity

A total of 920,715 good-quality sequences, averaging 50,591 sequences per sample, were obtained by analysing the intestine of *L. vannamei*. Venn diagram analysis showed that all groups had the same 169 operational taxonomic units (OTUs); the number of unique OTUs was the second highest in the HT and FM groups, and the lowest in the FF group (Figure 1A).

The alpha richness and plurality of intestinal microorganisms were evaluated using the Chao1, Simpson, and Shannon indices. Alpha diversity analysis showed that the Simpson and Shannon indices were significantly different among the groups (Table 2). The CK group had the lowest Chao1 index, and there were no significant differences among the experimental groups. In addition, the Shannon index of species diversity was lower in all the experimental groups compared to the HT group. There was no significant difference in the Simpson index between the FM and FF groups; however, the Simpson indices were similar and significantly higher in the CK and HT groups than in the FM and FF groups. The Shannon index of species diversity was lower in all the experimental groups than in the HT group. The Shannon index of species diversity was not significantly different between the FM and FF groups but was significantly lower in the HT group. The microbial samples from the FM and FF groups were clearly distinguishable based on the PCoA plot. The CK group could only be distinguished from the FM group and not from the other experimental groups (Figure 1B). 

### 3.3. Intestinal Microbial Composition 

Twenty bacterial phyla were identified in each group (Figure 2A). The bacterial composition of all groups consisted mainly of Proteobacteria, Bacteroidetes, Firmicutes, Tenericutes, and Verrucomicrobia. The bacterial composition of the intestine was dominated by Proteobacteria in all groups, with higher abundances in the FM (92.07%) and FF (91.07%) groups than in the CK group (70.33%). Notably, compared with the MC group, fly maggot supplementation decreased the relative abundance of Firmicutes.

At the class level, intestinal bacterial composition was dominated by Alphaproteobacteria and Gammaproteobacteria (Figure 2B). Gammaproteobacteria was more abundant in the FM group than in the CK group. The abundance of Alphaproteobacteria in the FF group (72.76%) was lower than that in the CK group, whereas the groups in the CK group were divergent with no common dominant species.

At the genus level, compared to the CK group, *Ruegeria* abundance in the FF group (51.67%) and *Vibrio* in the FM group (68.80%) increased, and *Pseudomonas*, *Antarctobacter*, *Nautella*, and *Lutimonas* were higher in the HT group. *Ruegeria* abundance was higher in the FF group (51.67%) than in the HT group (15.14%) (Figure 2C). 

### 3.4. Differential Analysis of Intestinal Microbiota

LEfSe was used to analyse the differential abundances of the three bacterial groups. Four bacterial taxa, *Clostridiaceae* (family), *Campylobacteraceae* (family), *Camplobacterales* (order), and *Epsilonproteobacteria* (family), were enriched in the FM group, three bacterial taxa, *Actinomycetales* (phylum), *Actinobacteria* (class), and *Actinobacteria* (phylum), were enriched in the FF group, and only one bacterial taxon, *Peptostreptococcaceae* (family), was enriched in the HT group (Figure 3A). Among the differential bacteria with LDA scores above 3.5, *Clostridiaceae* (family) dominated the FM group, whereas *Actinomycetales* (phylum), *Actinobacteria* (class), and *Actinobacteria* (phylum) dominated the FF group (Figure 3B).

### 3.5. The Correlation Network and Metabolic Analyses of Intestinal Microbiota

Microbial community interactions involve assemblages of microorganisms and their environments [29] that influence community stability and ecological characteristics [30]. In the bacterial phylum-based correlation network, Proteobacteria was negatively associated with Bacteroidetes, whereas Firmicutes was positively associated with Cyanobacteria. (Figure 4A). *Vibrio* and *Reugeria* had the highest abundance and were negatively correlated in the Bacteroidetes correlation network (Figure 4B).

### 3.6. Treatment with Fly Maggot Protein Could Affect the Microbiome Function

Based on the random forest KEGG classification, the predicted functions of the intestinal microflora of *L. vannamei* were analysed using the PICRUSt 2. Functional categories were similar across the groups. Analysis of level 2 and 3 biometabolism in the KEGG functional annotation revealed 20 subfunctional components (Figure 5A,B). 

Microbial metabolic functions affect *L. vannamei* growth. In this study, under the triple annotation depth of KEGG, the FF and HT groups showed some similarities in the metabolic functions of the intestinal microorganisms of shrimp after the intake of different treatments of fly maggot protein-fermented feeds. An analysis of microbial metabolic pathways in the intestine of shrimp in the FM group showed that there were significant differences between the shrimp that ingested fresh fly maggot protein and the remaining three groups in the above pathways. The metabolic pathways that were significantly upregulated were “beta-Lactam resistance”, “carbohydrate digestion and absorption”, “nitrotohuene degradation”, “NOD-like receptor signalling pathways”, “plant-pathogen interactions”, “renin-angiotensin system”, and “salmonella infection”, and “fluid shear stress and atherosclerosis” were significantly downregulated metabolic pathways. In contrast, the metabolic function of the intestinal microbes was not significantly altered in the FF and HT groups (Figure 5C). 

### 3.7. Correlations between Intestine Bacteria and the Growth of Shrimp

The correlation between microorganisms with weight gain rate (WG) and SR was analysed using correlation heat maps. SR was positively correlated with the abundance of *Lactobacillus*, *Microbulbifer*, *Rubellimicrobium*, *Marinoscillum*, and *Oscillospira* and negatively correlated with *Glaciecola*, *Hahella*, and *Bacteriovorax* abundance. WG positively correlated with *Burkholderia* abundance and negatively correlated with *Acidaminobacter* abundance (Figure 6).

## 4. Discussion

The application of fly maggot protein in aquatic animals has been reported to enhance the survival and growth of aquatic animals and improve intestinal microbiota. However, the effects of fly maggot protein from different treatment methods on aquatic animals still need to be investigated. In the present study, high-temperature pelleted fly maggot protein-fermented feed increased the survival and weight gain rates of *L. vannamei* significantly. Regarding the application of fly maggot protein on *L. vannamei*, existing studies have also shown that fly maggot protein can improve the SR of *L. vannamei* [7]. However, in this study, although fly maggot protein was present in all the feeds of the experimental groups, the results obtained from the different treatments differed. In the present study, fly maggot protein was fermented at high temperatures after fermentation, ensuring that the same amount of fly maggot protein was supplied to the diet. Zhang et al. [31] reported that fermented feed promoted growth. The WG of shrimp fed either fly maggot protein-fermented feed or high-temperature pelleted fly maggot protein-fermented feed was significantly higher than that of the shrimp that were fed fresh fly maggot protein. This phenomenon suggests that fly maggot protein can significantly promote shrimp growth after fermentation and high-temperature treatments.

The functional activity and stability of the intestinal microbiota are important for shrimp health because they perform many functions related to immunity and pathogen resistance [32]. Previous studies have shown that the addition of fly maggot protein powder to fattening pig feed can inhibit the growth of harmful intestinal bacteria, promote the proliferation of beneficial bacteria, and effectively improve the digestion and absorption of nutrients [33]. These studies indicate that fly maggot proteins can alter the composition of intestinal microorganisms. Similarly, fly maggot proteins altered the relative abundance of the dominant intestinal bacteria in *L. vannamei* in this study, including an increase in the level of Proteobacteria in all experimental groups, Bacteroidetes in the HT group, and a decrease in the level of Firmicutes in all experimental groups. These results suggest that the changes in the relative abundance of the dominant bacteria produced by the fly maggot Proteobacteria varied among the treatments.

Some potentially beneficial bacterial genera, especially health-related host bacteria, differed between the groups. One group of potential probiotics is *Roseobacter*, which produces tropodithietic acid, which can kill or reduce the growth of several *Vibrio* pathogens in multiple aquaculture systems [34,35,36,37]. Kang et al. [38] found that *Pseudomonas aeruginosa* UCBPP-PA14 significantly lyses *Microcystis aeruginosa* and degrades microcystins. In this study, after the ingestion of fermented and high-temperature fly maggot protein feed by the shrimp, the relative abundances of *Ruegeria* and *Pseudomonas* were elevated, suggesting that fermented fly maggot protein feed can inhibit the growth of harmful bacteria by enhancing the abundance of beneficial bacteria and inhibiting the growth of harmful bacteria. In contrast, certain potentially harmful bacterial genera differed between the groups. *Vibrio* is a Gram-negative bacterium, and *Vibrio parahaemolyticus* infestation increases intestinal permeability and inhibits glucose and amino acid uptake by *L. vannamei* [39]. In the present study, the relative abundance of *Vibrio* was elevated in the FM group, suggesting that the ingestion of fresh fly maggot protein by shrimp may disrupt intestinal cell membrane permeability and be detrimental to nutrient uptake, thereby inhibiting shrimp growth. In addition, based on the bacterial correlation network, there was a positive correlation between the beneficial bacteria mentioned above, whereas *Vibrio*, a harmful bacterium, was negatively correlated with the beneficial bacteria. This was similar to the conclusions of Faust et al. [26]. Bacteria may cooperate to build biofilms that lead to antibiotic resistance among their members, resulting in mutually beneficial symbiotic relationships; conversely, negative interactions occur when metabolic by-products of one microorganism alter the environment to the detriment of other microorganisms [26]. Therefore, it is predicted that fly maggot protein-fermented feed can maintain the homeostasis of beneficial bacteria in the intestine of *L. vannamei*, which is beneficial for intestinal health. However, fresh fly maggot proteins alter the intestinal microbiome and disrupt the symbiotic relationship between the bacteria, resulting in the invasion of *Vibrio* and other pathogenic bacteria that may damage the intestine and affect shrimp health.

The metabolic functions of microorganisms influence *L. vannamei* growth. In this study, the results of KEGG pathway analysis based on the depth of triple annotation showed that significant differences in intestinal microbial metabolic functions occurred after the shrimp were fed different fly maggot protein diets. In this study, we found that the fly maggot protein significantly enhanced beta-lactam resistance in the intestinal microorganisms of *L. vannamei*. Beta-lactam antibiotics are widely used to inhibit bacterial growth by inhibiting the synthesis of bacterial cell walls [40] and destroying bacterial structure [41]. Antibiotic resistance genes are considered novel environmental contaminants [42] with a long-lasting high transmission frequency and harmfulness. In contrast, the beta-lactam resistance of the intestinal microorganisms was significantly reduced after feeding with fly maggot protein-fermented feed and fermented feed pelleted at high temperatures. Therefore, feeding fresh fly maggot proteins increases the abundance of antibiotic-resistant bacteria in the intestines of *L. vannamei*, which is detrimental to human health and ecological safety. In contrast, fermented feed significantly reduced the relative abundance of antibiotic-resistant bacteria in the shrimp, thereby mitigating environmental harm.

The intestinal tract is inhabited by numerous microorganisms that affect the host’s digestion and immunity. Stable microbial communities can promote host health by producing beneficial metabolites and immune stimulation [19]. In the present study, the growth of *L. vannamei* was assessed based on survival and WG rates. Beneficial bacterial genera play an important role in the healthy growth of shrimp. We found that the relative abundances of *Lactobacillus*, *Micrococcus*, *Erythrobacter*, *Marinoscillum*, and *Oscillospira* were significantly and positively correlated with survival rates. For example, lactic acid produced by *Lactobacillus*, a probiotic in fermented feeds, is an osmotic agent for the outer membrane of Gram-negative pathogens and increases the susceptibility of pathogenic bacteria to antimicrobial molecules [43]; on the other hand, the culture extracts of *Microbulbifer* also exhibit unique broad-spectrum antimicrobial activity against Gram-positive and Gram-negative bacteria and fungi [44]. Second, *Rubellimobium* spp. are representatives of the *Roseobacter* clade in Rhodobacteraceae [45], and as a probiotic for aquaculture, *Roseobacter* maintains its anti-pathogenic activity against pathogenic aquaculture cultures (live feeds and fish eggs/larvae) in the live antagonistic activity of *Vibrio vulnificus* [37]. Finally, *Oscillospira* produces short-chain fatty acids (SCFAs), which have been identified as next-generation probiotic candidates because of their positive effects on specific diseases, such as metabolic disorders related to obesity [46]. On the other hand, *Marinoscillum*, a member of the family “Flexibacteraceae” and the phylum “Bacteroidetes” [47,48], can inhibit the growth of *Vibrio* [49,50]. In contrast, *Glaciecola*, *Hahella*, and *Bacteriovorax* were significantly negatively correlated with survival. *Glaciecola* and *Hahella* are members of the “Gammaprotobacteria” class [51,52], but due to the production of metabolites that can specifically kill harmful algae [53], the role of *Hahella* in aquaculture requires further analysis. In addition, *Bacteriovorax* is a member of the phylum “Proteobacteria” based on the similarity of affinities [54]. The negative correlation between *Bacteriovorax* and shrimp survival is also consistent with the finding that Proteobacteria often act as the dominant bacteria in diseased shrimp [55].

*Burkholderia* was positively correlated with WG, and members of *Burkholderia* have both benefits and drawbacks, first as plant and human pathogens [56,57], but also through their own enzymes to degrade certain contaminants [58]. Therefore, the role of *Burkholderia* in *L. vannamei* needs to be analysed in greater detail. Based on the results of this study, *Burkholderia cepacia* has growth-promoting effects. The abundance of *Acidaminobacter* was negatively correlated with WG. *Acidaminobacter* is a member of Gammaproteobacteria [59], which includes pathogenic bacteria in aquaculture environments, such as *Vibrio*, which are detrimental to shrimp health and growth.

These findings also corroborate that probiotic bacteria are positively correlated with the survival and weight gain rates of the bacteria, whereas harmful bacteria are negatively correlated with the survival and weight gain rates of *L. vannamei*. In addition, by combining the intestinal microbial composition of each group, we found that most probiotic bacteria, such as *Roseobacter*, were present in the FF and HT groups, suggesting that protein-fermented fly maggot feed increased the relative abundance of beneficial bacteria in the intestinal tract of *L. vannamei*, which was beneficial to the health of the hosts. In contrast, most harmful bacteria, such as *Vibrio*, were from the FM group, which also suggests that fresh fly maggot protein can disrupt the homeostasis of the intestinal tract of shrimp, resulting in an increase in the abundance of harmful bacteria, thus jeopardising the health of shrimp. In addition, the abundance of *Vibrio anguillarum* in shrimp intestines increases linearly with the progression of acute hepatopancreatic necrotic disease [60], suggesting that fresh fly maggot proteins lead to an increase in the number of harmful bacteria, which may be detrimental to shrimp health. However, there are some limitations to this study. For example, the intestinal gut structure was not investigated, and the effects of fly maggot proteins produced in the intestines under different treatments were not analysed from a physiological point of view. In addition, gene expression and enzyme activities related to the healthy growth and development of *L. vannamei* were not explored in this study. Therefore, future studies should analyse the effects of fly maggot proteins on *L. vannamei* under different treatments from multiple perspectives, including gene expression, enzyme activity, and histology.

## 5. Conclusions

Overall, we found that fly maggot protein improved the growth and survival of *L. vannamei* after fermentation and high-temperature treatment. In addition, different treatments with fly maggot protein altered the relative abundance of dominant bacteria, particularly beneficial substance-producing bacteria (*Ruegeria* and *Pseudomonas*) and pathogenic bacteria (*Vibrio*). Different treatments with fly maggot protein also affected the metabolic functions of microorganisms in the shrimp intestines, and the resulting differences in the relative abundance of beneficial and harmful intestinal microorganisms were significantly correlated with weight gain and survival rates. These results suggest that differently treated fly maggot proteins have different effects on the microbial composition of the intestinal tract of *L. vannamei*, and rebuilt protein-fermented feeds are ideal for use in the context of feeding shrimp in a commercial setting.

## Figures and Tables

**Figure 1 biology-12-01433-f001:**
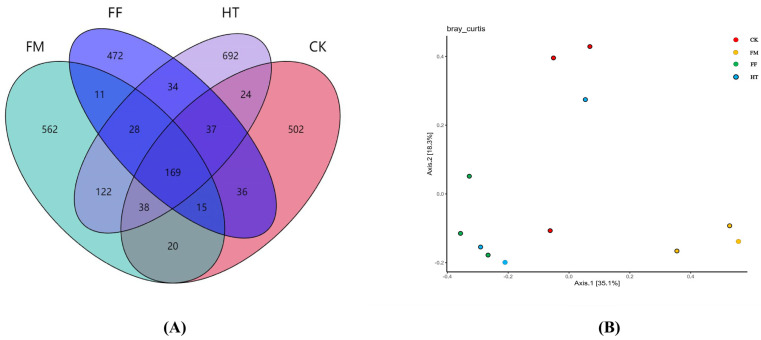
Diversity and richness of intestinal microbial variability in *Litopenaeus vannamei*. (**A**) Venn diagram; (**B**) PCoA analysis on the basis of Bray-Curtis distance.

**Figure 2 biology-12-01433-f002:**
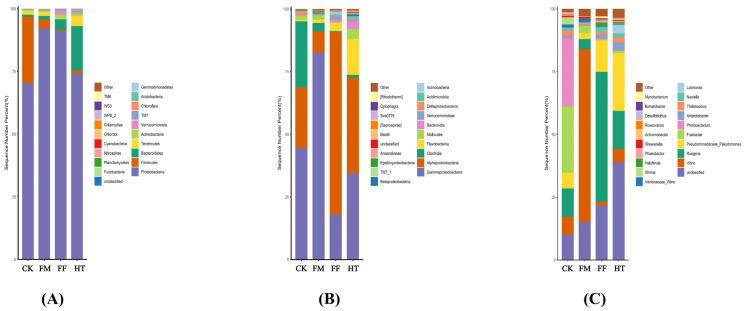
Changes in the composition of the intestinal microbial community of *Litopenaeus vannamei*. (**A**) relative bacterial abundance at the phylum level; (**B**) relative bacterial abundance at the class level; (**C**) relative bacterial abundance at the genus level.

**Figure 3 biology-12-01433-f003:**
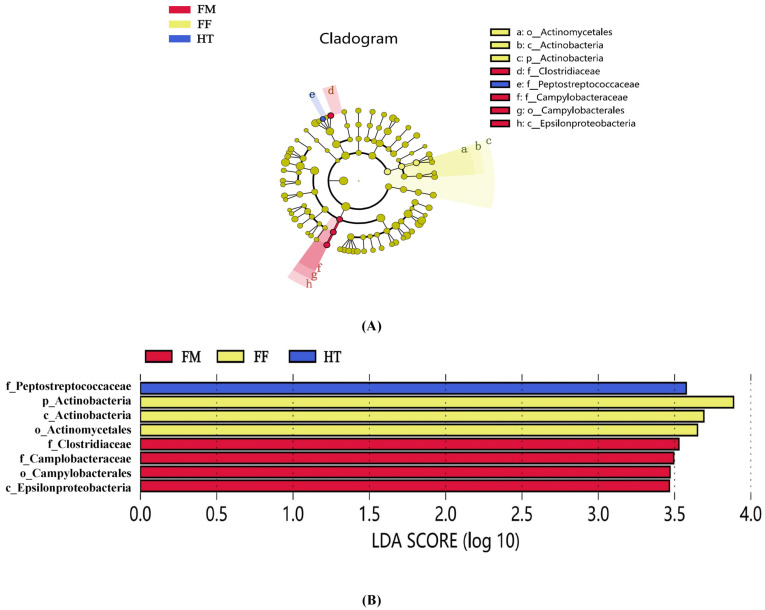
Between-group variability of intestinal microorganisms of *Litopenaeus vannamei*. (**A**) In the LEfSe cladogram, the diameter of each circle is proportional to the bacterial taxon abundance. (**B**) LDA scores of LEfSe-PICRUSt, only the taxon with LDA values > 3.5 are shown. The lowercase letters in front of the strain name represent the taxonomic level at which the strain is located: k for kingdom, p for phylum, c for class, o for order, f for family, g for genus, and s for species.

**Figure 4 biology-12-01433-f004:**
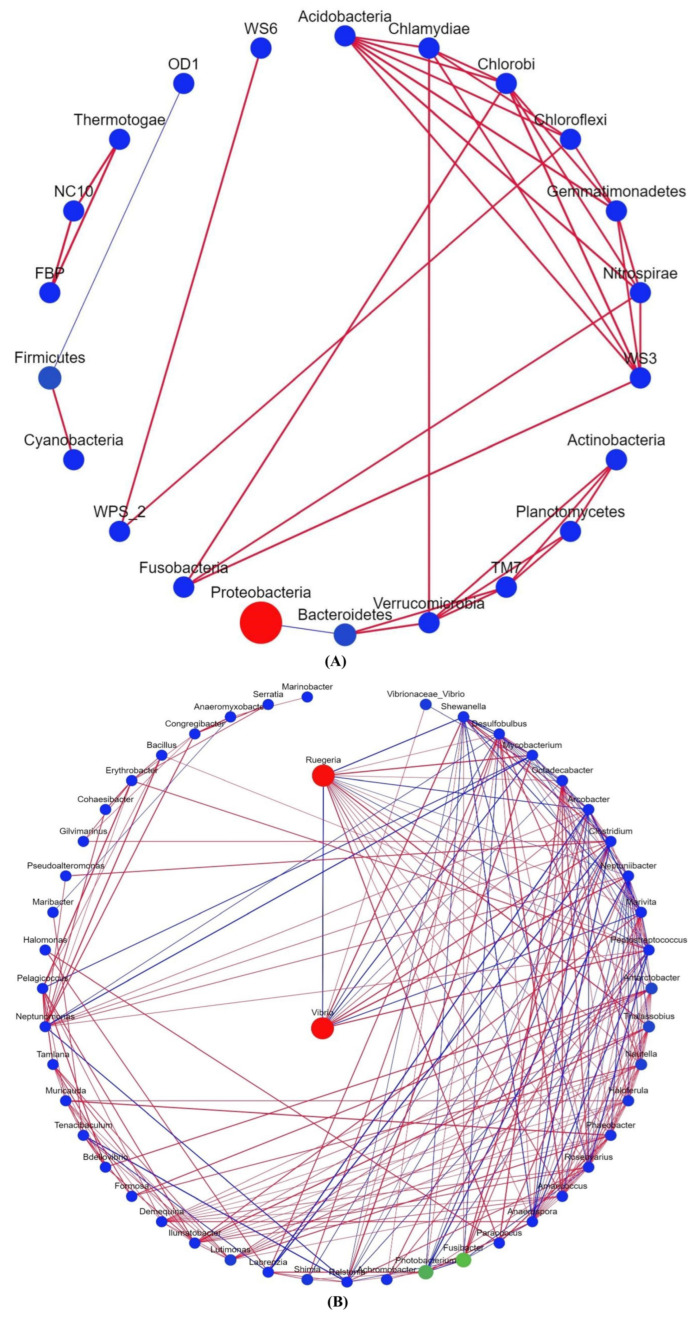
Correlation network analyses of intestinal microbial of *Litopenaeus vannamei*. (**A**) The correlation network based on bacterial phyla. (**B**) The correlation network of bacteria genera. Circles indicate the species, size indicates its relative abundance, different colours indicate different species phylum classifications, the lines connecting the circles denote the significant correlation between the two species (*p* < 0.05), red lines represent positive correlations, and blue lines represent negative correlations. The thicker the line, the greater the absolute value of the correlation coefficient.

**Figure 5 biology-12-01433-f005:**
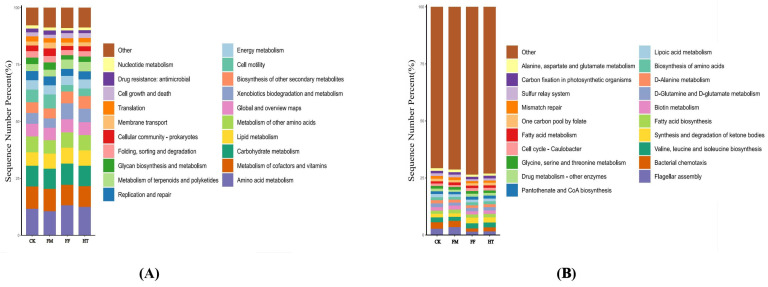
Analysis of bacterial metabolism in the intestinal KEGG pathway of *Litopenaeus vannamei* based on fermented feed and fly maggot protein: (**A**) results of KEGG level 2 analysis; (**B**) results of KEGG level 3 analysis; (**C**) results of the predicted variability of the pathway groups under KEGG level 3. Different letters are used to show significant differences (*p* < 0.05).

**Figure 6 biology-12-01433-f006:**
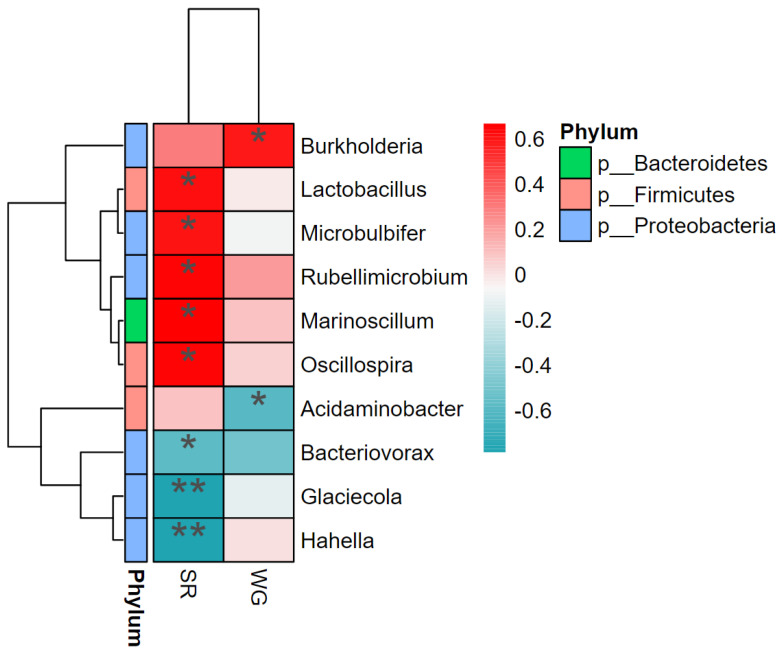
Heat map of the interrelationship between microbial species and environmental factors (SR and WG) at the genus level. * Denotes significant correlation (**p* < 0.05; ***p* < 0.01).

**Table 1 biology-12-01433-t001:** Changes in the growth performance and survival rate of *Litopenaeus vannamei*.

Items	CK	FM	FF	HT
Initial weight/g	3.95 ± 0.20 ^a^	4.14 ± 0.18 ^a^	3.93 ± 0.14 ^a^	3.85 ± 0.089 ^a^
Final weight/g	10.07 ± 0.06 ^b^	11.38 ± 0.36 ^a^	8.32 ± 0.58 ^c^	12.19 ± 0.35 ^a^
Specific growth rate/%	3.47 ± 0.17 ^b^	3.75 ± 0.13 ^ab^	2.77 ± 0.36 ^c^	4.27 ± 0.02 ^a^
Weight gain rate/%	155.60 ± 11.84 ^b^	113.00 ± 20.48 ^c^	175.80 ± 9.31 ^ab^	216.60 ±1.96 ^a^
Survival rate/%	75.33 ± 6.43 ^a^	69.33 ± 5.03 ^a^	75.33 ± 10.06 ^a^	77.33 ± 4.16 ^a^
Mortality rate/%	24.67 ± 6.43 ^a^	30.67 ± 5.03 ^a^	24.67 ± 10.06 ^a^	22.67 ± 4.16 ^a^

Note: The different lowercase letters indicate a significant difference (*p* < 0.05).

**Table 2 biology-12-01433-t002:** Changes in intestinal microbial alpha-diversity of *Litopenaeus vannamei*.

Items	CK	FM	FF	HT
Chao Index	556.00 ± 140.00 ^a^	293.00 ± 108.00 ^a^	673.00 ± 330.00 ^a^	530.00 ± 219.00 ^a^
Shannon Index	5.84 ± 0.28 ^b^	2.30 ± 0.71 ^a^	3.00 ± 0.58 ^a^	5.720 ± 0.34 ^b^
Simpson	0.95 ± 0.01 ^b^	0.49 ± 0.12 ^a^	0.54 ± 0.13 ^a^	0.97 ± 0.01 ^b^

Note: The different lowercase letters indicate a significant difference (*p <* 0.05).

## Data Availability

Data are contained within the article and supplementary materials.

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
