# Peer review of "Effect of Fly Maggot Protein as Dietary on Growth and Intestinal Microbial Community of Pacific White Shrimp *Litopenaeus vannamei"

_biology, 2023, doi:10.3390/biology12111433_

Round 1
Reviewer 1 Report
Comments and Suggestions for Authors
Comments to the Author
· 1. Please add a table for diet preparation and ingredients and amino acid composition.
2. The authors should mention in detail how they deal with fly maggot protein. For example, how did they extract protein from fly maggots? what methods of fermentation did they use???
3. Lake of the novelty of the research study as there's now a new idea or methodology in this research study.
https://doi.org/10.1016/j.aquaculture.2017.02.022
https://www.cabdirect.org/cabdirect/abstract/20123245407
DOI: 10.3724/SP.J.1231.2012.27621
DOI:10.3390/ani11051450
DOI: 10.3724/SP.J.00001
Cheng X, Chen Z, Yan Z, Zhao P, Wang F, Li G. Effects of housefly protein on growth performance, immunity and muscular composition in Pacific white leg shrimp Litopenaeus vannamei. Fisheries Science (Dalian). 2018;37(3):324-9.
4. Histopathology of the intestine is very important in this study. Why the authors didn’t apply histopathology?
· 5. After the 30-day feeding trial?? Is this enough? The feeding trial should be 56 days or more to wisely evaluate the effect of fly maggot protein on growth performance.
· 6. Add reference for the growth performance and feed utilization efficiency formula.
· 7. How did the authors check the DNA purity?
· 8. Authors must be discussing your result; this type of discussion is not reasonable and not solid for a real discussion. It seems that every favorable result is a result of boosting effects of the supplement and any unfavorable ones are results of reverse effects! In this way, all the effects could be explained
· 9. The authors should mention the limitations of their findings and their perspectives.
· 10. Please improve your conclusion it's poorly written.
·
Comments on the Quality of English LanguageThe English quality is very poor. Many grammatical and spelling errors are present in the manuscript. Extensive editing of the English language and style is required.
Author Response
Dear reviewer,
Thanks for the comments on the manuscript. Please find attached the revised manuscript. We have carefully modified the manuscript by considering the referees’ comments. Please see "Response to Reviewers' Comments" for more details. In addition, some other minor revisions have been performed to improve readability and the layout of the manuscript. Moreover, the related portions are highlighted in bule color.
In closing, I should like to thank you for the valuable and detailed comments.
Sincerely yours,
Reply to the Referees’ comments:
Reviewers #1:
- Please add a table for diet preparation and ingredients and amino acid composition.
[Reply] We apologize that we did not give the feed formulation and the table of ingredients and amino acid composition in the manuscript before; the feed formulation and ingredients have been added in the supplementary information. Lines 143-162 of the manuscript.
- The authors should mention in detail how they deal with fly maggot protein. For example, how did they extract protein from fly maggots? what methods of fermentation did they use???
[Reply] Thank you very much for your suggestion. Fresh fly maggots are cleaned and decontaminated, sterilized and cleaned using an ozone cell, and then grinded. Their company does not extract protein from the fly maggots, they just treat them as described above. Manuscript lines 143 to 145.
- Lake of the novelty of the research study as there's now a new idea or methodology in this research study.
https://doi.org/10.1016/j.aquaculture.2017.02.022
https://www.cabdirect.org/cabdirect/abstract/20123245407
DOI: 10.3724/SP.J.1231.2012.27621
DOI:10.3390/ani11051450
DOI: 10.3724/SP.J.00001
Cheng X, Chen Z, Yan Z, Zhao P, Wang F, Li G. Effects of housefly protein on growth performance, immunity and muscular composition in Pacific white leg shrimp Litopenaeus vannamei. Fisheries Science (Dalian). 2018;37(3):324-9.
[Reply] Thank you very much for your guidance. In this study, we investigated the effects of different treatments of fly maggot protein on Litopenaeus vannamei through changes in intestinal microorganisms. The results of other papers were referred to in terms of subjects and feed ingredients, but this study wanted to analyze the effects of fly maggot protein through the perspective of intestinal microorganisms, so as to fill the gap in this area of research. Therefore, we apologize for not analyzing the results of the experiment by other means, as the experiment was designed to focus only on this issue.
- Histopathology of the intestine is very important in this study. Why the authors didn’t apply histopathology?
[Reply] We apologize that we did not process the intestines for sectioning and observation. This is a problem with our experiment.
- After the 30-day feeding trial?? Is this enough? The feeding trial should be 56 days or more to wisely evaluate the effect of fly maggot protein on growth performance.
[Reply] We apologize, we had problems with the writing and the farming time was 8 weeks instead of 30 days. Line 171 in the manuscript.
- Add reference for the growth performance and feed utilization efficiency formula.
[Reply] Thank you very much for your suggestion, we have added reference to the above two formulas. Lines 182 to 184 of the manuscript.
- How did the authors check the DNA purity?
[Reply] We used Qubit to detect DNA concentration.
- Authors must be discussing your result; this type of discussion is not reasonable and not solid for a real discussion. It seems that every favorable result is a result of boosting effects of the supplement and any unfavorable ones are results of reverse effects! In this way, all the effects could be explained
[Reply] Thank you very much for your suggestion. We have revised the discussion section after reworking. Lines 411 to 414 in the manuscript as well as line 476 to 479 in the manuscript.
- The authors should mention the limitations of their findings and their perspectives.
[Reply] Thank you very much for your suggestion. We have analyzed the limitations of this study at the end of the discussion. Lines 479 to 486 of the manuscript.
- Please improve your conclusion it's poorly written.
[Reply] Thank you very much for your suggestion, we have optimized the conclusion section. Lines 488 to 498 of the manuscript.
Please see the attachment

Reviewer 2 Report
Comments and Suggestions for Authors
In this study, the authors included fly maggot meal in the diets of white shrimp and the growth and intestinal microbiota were measured.
There are two serious flaws here. First, there is no description of how the diets were made or formulated. Just that is reason enough to reject. It is imperative to show how the diets were formulated with ingredients for each diet, and then to conduct proximate composition on them. This was not done, and therefore the rest of the results cannot be validated. Second, growth trials should last for at least 60 days and not 30 days as it was done here. The survival of the shrimp after 30 days was also quite low and this gives reason to question the condition the shrimp were cultured in.
Comments on the Quality of English LanguageMostly fine
Author Response
Dear reviewer,
Thanks for the comments on the manuscript. Please find attached the revised manuscript. We have carefully modified the manuscript by considering the referees’ comments. Please see "Response to Reviewers' Comments" for more details. In addition, some other minor revisions have been performed to improve readability and the layout of the manuscript. Moreover, the related portions are highlighted in bule color.
In closing, I should like to thank you for the valuable and detailed comments.
Sincerely yours,
Reply to the Referees’ comments:
In this study, the authors included fly maggot meal in the diets of white shrimp and the growth and intestinal microbiota were measured.
There are two serious flaws here. First, there is no description of how the diets were made or formulated. Just that is reason enough to reject. It is imperative to show how the diets were formulated with ingredients for each diet, and then to conduct proximate composition on them. This was not done, and therefore the rest of the results cannot be validated. Second, growth trials should last for at least 60 days and not 30 days as it was done here. The survival of the shrimp after 30 days was also quite low and this gives reason to question the condition the shrimp were cultured in.
[Reply] Thank you very much for pointing this out. We apologize for the error of not stating how the feed was made and the ingredients, we have added the feed ingredients in the supplementary material, lines 143-162 in the manuscript; we are very sorry for the error in the statistics of the number of days of breeding, our number of days of breeding should have been 8 weeks, line 171 in the manuscript. For the aspect of lower survival rate, we followed strict breeding practices during the breeding process and we are very sorry for the survival rate results.
Please see the attachment

Reviewer 3 Report
Comments and Suggestions for Authors
Comments to author
Manuscript Number/id- biology-2662485
The manuscript entitled “Effect of dietary fly maggot protein on growth and intestinal microbial community of Litopenaeus vannamei” is a nice piece of work done by the authors. In this manuscript, the author reports the impact of different treatments of fly maggot protein feed on the growth and intestinal flora of Pacific white shrimp (Litopenaeus vannamei). Fresh fly maggot protein negatively affects shrimp growth, while fermented and high-temperature pelleted feeds improve survival. These treatments increase beneficial bacteria in the intestines and reduce harmful ones, while fresh protein can invade Vibrio and antibiotic-resistant bacteria. This research supports the use of fly maggot protein in aquaculture.
The topic is of interest, and the manuscript is well illustrated. However, the authors need to respond to some comments/suggestions
Minor Correction required
The title could be revised to
“Effect fly maggot protein as dietary on growth and intestinal microbial community of Pacific white shrimp Litopenaeus vannamei (Boone, 1931)”
In the introduction section
Line 112-113 Shrimp growth is influenced by a number of factors, including food type and intestinal microbial. Change into Shrimp growth is influenced by a number of factors, including food type and intestinal microbiota
Material and methods
Line 133 Authors may write ‘A total of 720 numbers of….’
Line 134 Reduce the ±SD value to three decimals
Line 137 As shrimps are susceptible to Dissolved Oxygen (DO), authors may highlight the range of DO maintained during culture
Line 147 It is suggested that a short storage procedure of the feed may be added
Line 149 Authors may clearly write the full form of the abbreviated ‘CK,FM,FF,HT ‘
Line 157 Authors may rewrite the sentence
Line 160-161 The meaning has not been addressed clearly as why the 5 samples of intestines were mixed.
Results
Table 1 Authors may include the % mortality occurred in each treatment
Table 1 & 2. The letters ‘a’ ‘b’ should be superscript
Line 244 Authors may include the variance (%) in 30FM and FF groups if possible
Discussion
Line 388 This is also similar to the conclusion reached by Faust et al. (mention the year)
Line 394 Change intestinal to intestine
The authors seem to have carried out the work on the reference level of crude protein content (%) of the fly. Clarification is also required on the crude protein content (%) of the present test group CK,FM,FF and HT if carried out before the start of the experiment. If not, then justification may be added in the discussion
Comments on the Quality of English LanguageMinor editing of English language required
Author Response
Dear reviewer,
Thanks for the comments on the manuscript. Please find attached the revised manuscript. We have carefully modified the manuscript by considering the referees’ comments. Please see "Response to Reviewers' Comments" for more details. In addition, some other minor revisions have been performed to improve readability and the layout of the manuscript. Moreover, the related portions are highlighted in bule color.
In closing, I should like to thank you for the valuable and detailed comments.
Sincerely yours,
Reply to the Referees’ comments:
Manuscript Number/id- biology-2662485
The manuscript entitled “Effect of dietary fly maggot protein on growth and intestinal microbial community of Litopenaeus vannamei” is a nice piece of work done by the authors. In this manuscript, the author reports the impact of different treatments of fly maggot protein feed on the growth and intestinal flora of Pacific white shrimp (Litopenaeus vannamei). Fresh fly maggot protein negatively affects shrimp growth, while fermented and high-temperature pelleted feeds improve survival. These treatments increase beneficial bacteria in the intestines and reduce harmful ones, while fresh protein can invade Vibrio and antibiotic-resistant bacteria. This research supports the use of fly maggot protein in aquaculture.
The topic is of interest, and the manuscript is well illustrated. However, the authors need to respond to some comments/suggestions
[Reply] Thanks for the positive comments.
Minor Correction required
- The title could be revised to
“Effect fly maggot protein as dietary on growth and intestinal microbial community of Pacific white shrimp Litopenaeus vannamei (Boone, 1931)”
[Reply] Thank you very much for your suggestion. We have made changes to the title.
In the introduction section
- Line 112-113 Shrimp growth is influenced by a number of factors, including food type and intestinal microbial. Change into Shrimp growth is influenced by a number of factors, including food type and intestinal microbiota
[Reply] Thank you very much for your suggestion, we have changed it accordingly. Manuscript lines 113 to 114.
Material and methods
- Line 133 Authors may write ‘A total of 720 numbers of….’
[Reply] Thank you for your suggestion, which has been modified accordingly. Line 134 of the manuscript.
- Line 134 Reduce the ±SD value to three decimals
[Reply] Thank you very much for your suggestion, which has been modified accordingly. Line 135 of the manuscript.
- Line 137 As shrimps are susceptible to Dissolved Oxygen (DO), authors may highlight the range of DO maintained during culture
[Reply] Thank you very much for pointing out that we add to the dissolved oxygen range. Lines 138 to 139 of the manuscript.
- Line 147 It is suggested that a short storage procedure of the feed may be added
[Reply] Thank you very much for your suggestion, which has been modified accordingly. Lines 162 to 164 of the manuscript.
- Line 149 Authors may clearly write the full form of the abbreviated ‘CK, FM, FF, HT ‘.
[Reply] Thank you very much for your suggestions, changes have been made to the group abbreviations. Line 167 of the manuscript.
- Line 157 Authors may rewrite the sentence
[Reply] Thank you very much for your suggestion, it has been revised accordingly. Lines 173 to 174 of the manuscript.
- Line 160-161 The meaning has not been addressed clearly as why the 5 samples of intestines were mixed.
[Reply] Thank you very much for pointing out that we mixed 5 intestinal samples after considering the inter-individual differences. According to the existing research, the more samples of intestine, the less inter-individual variation affects the experimental results, so we chose to mix 5 samples of intestine. Lines 176 to 178 in the manuscript.
Results
- Table 1 Authors may include the % mortality occurred in each treatment
[Reply] Thank you very much for your suggestion, the addition of mortality rates to the groups has been completed.
Table 1 & 2. The letters ‘a’ ‘b’ should be superscript
[Reply] Thank you very much for your suggestions, the letters in Tables 1 and 2 are superscripted.
- Line 244 Authors may include the variance (%) in 30FM and FF groups if possible
[Reply] Thank you very much for your suggestion, the differences between the FM and FF groups have been listed, lines 262 to 264 in the manuscript.
Discussion
- Line 388 This is also similar to the conclusion reached by Faust et al. (mention the year)
[Reply] Thank you very much for the reminder, the cited literature for this conclusion has been added. Line 405 of the manuscript.
- Line 394 Change intestinal to intestine
[Reply] Thank you very much for your suggestions, which have been revised accordingly. Line 410 of the manuscript.
- The authors seem to have carried out the work on the reference level of crude protein content (%) of the fly. Clarification is also required on the crude protein content (%) of the present test group CK, FM, FF and HT if carried out before the start of the experiment. If not, then justification may be added in the discussion.
[Reply] Thank you very much for your suggestion, we have added the formula, composition and amino acid content of the feed for each group. The point of difference from the CK group is that the experimental group used fly maggot protein as a protein source and did not use fish meal. Lines 143 to 162 in the manuscript.
Please see the attachment

Round 2
Reviewer 1 Report
Comments and Suggestions for Authors
Thank you so much for your response, however, the English quality is inferior. Many grammatical and spelling errors are present in the manuscript. Extensive editing of the English language and style is required.
Comments on the Quality of English LanguageEnglish editing is still required
Author Response
Dear reviewer,
Thanks for the comments on the manuscript. Please find attached the revised manuscript. We have carefully modified the manuscript by considering the referees’ comments. The details are described in “Reply to the Referees’ comments” on next pages. In addition, some other minor revisions have been performed to improve readability and the layout of the manuscript. Moreover, the related portions are highlighted in bule color.
In closing, I should like to thank you for the valuable and detailed comments.
Sincerely yours,
Reply to the Referees’ comments
Thank you so much for your response, however, the English quality is inferior. Many grammatical and spelling errors are present in the manuscript. Extensive editing of the English language and style is required.
[Reply] Thank you very much for your suggestion, we have re-touched up the article and have already made detailed corrections for incorrect grammar and spelling.

Reviewer 2 Report
Comments and Suggestions for Authors
Insufficient answers -- still recommend reject
Author Response
Dear reviewer,
Thank you for your comments on the manuscript. A revised version is attached. We apologize for not responding adequately to the last revision of the manuscript, and we have responded to your comments in detail and revised them carefully for this revision. We have carefully revised the manuscript taking into account the reviewers' comments. Please see "Response to Reviewers' Comments" on the next page for details.
Finally, thank you for your valuable and detailed comments.
Best regards,
Response to Reviewers' Comments
Round 1
In this study, the authors included fly maggot meal in the diets of white shrimp and the growth and intestinal microbiota were measured.
There are two serious flaws here. First, there is no description of how the diets were made or formulated. Just that is reason enough to reject. It is imperative to show how the diets were formulated with ingredients for each diet, and then to conduct proximate composition on them. This was not done, and therefore the rest of the results cannot be validated. Second, growth trials should last for at least 60 days and not 30 days as it was done here. The survival of the shrimp after 30 days was also quite low and this gives reason to question the condition the shrimp were cultured in.
[Reply] Thank you very much for pointing this out. The feed was prepared as follows, while fish meal was used as the protein source for ordinary compound feeds, fly maggot protein was selected as the protein source for the experimental feeds on the basis of not adding fish meal. Fresh fly maggots were cleaned, decontaminated, sterilized, ozonated in the ozone cell and milled to obtain the fly maggot protein source (BoYuJia Biotechnology, ShenZhen, China). All the test feeds were made without fish meal, 30% of fresh fly maggot protein was mixed into the normal feed without fish meal, 30% of fresh fly maggot protein was added into the fly maggot protein fermented feed and then fermented for 7 days without high temperature pelleting, while 30% of fresh fly maggot protein was added into the fly maggot protein pellet feed and then fermented for 7 days before pelleting at high temperature. Lines 144-151 in the manuscript. We apologize for not describing the method of preparation and composition of the feeds. We have added the feed ingredients, amino acid composition, and feed formulation to the supplemental material and added them below, with detailed revisions in lines 151-152 of the manuscript. The feed ingredients were determined as follows: Each treatment was tested in triplicate to obtain the average value, and the routine nutrient composition of the feeds was tested using the AOAC method. Among them, crude protein content was determined using a Kjeldahl nitrogen meter (BUCHI, KjeIIex K-360, Switzerland); crude fibre content was determined from the weight of dried residue after extracting the feed with acid and alkali; calcium ion content was determined using disodium ethylenediaminetetraacetic acid complexometric titration; phosphorus content was determined using spectrophotometry; and crude ash content was determined using scorching in a Marfo oven at 550 ℃. Moisture content was determined using the constant temperature drying method at 105 ℃. And the amino acid composition was determined as follows: Methionine and cystine were hydrolysed using oxidative acid hydrolysis, and the rest of the amino acids were hydrolysed by acid hydrolysis, with detailed revisions in lines 153-162 of the manuscript. we are very sorry for the error in the statistics of the number of days of breeding, our number of days of breeding should have been eight weeks, line 172 in the manuscript. For the aspect of lower survival rate, we followed strict breeding practices during the breeding process and we are very sorry for the survival rate results.
Table S1. Feed formulation for each group. (A) Feed composition of the CK group;(B) Feed composition of the FM group; (C) Feed composition of the FF group; (D) Feed composition of the HT group, unit: %
|
feed ingredients |
CK |
|
fishmeal |
30 |
|
soybean meal (processed GMO soybeans) |
23 |
|
peanut bran |
15 |
|
yeast |
5 |
|
flour |
20.3 |
|
soy lecithin |
1 |
|
fish oil |
1 |
|
soybean oil |
1 |
|
choline chloride |
0.6 |
|
potassium dihydrogen phosphate |
1 |
|
multivitamin |
1 |
|
multimineral |
1 |
|
Vc Phosphatidic Acid |
0.1 |
(A)
|
feed ingredients |
FM |
|
fly maggot protein |
30 |
|
soybean meal (processed GMO soybeans) |
23 |
|
peanut bran |
15 |
|
yeast |
5 |
|
flour |
20.3 |
|
soy lecithin |
1 |
|
fish oil |
1 |
|
soybean oil |
1 |
|
choline chloride |
0.6 |
|
potassium dihydrogen phosphate |
1 |
|
multivitamin |
1 |
|
multimineral |
1 |
|
Vc Phosphatidic Acid |
0.1 |
(B)
|
feed ingredients |
FF |
|
fly maggot protein |
30 |
|
soybean meal (processed GMO soybeans) |
23 |
|
peanut bran |
15 |
|
yeast |
5 |
|
flour |
20.3 |
|
soy lecithin |
1 |
|
fish oil |
1 |
|
soybean oil |
1 |
|
choline chloride |
0.6 |
|
potassium dihydrogen phosphate |
1 |
|
multivitamin |
1 |
|
multimineral |
1 |
|
Vc Phosphatidic Acid |
0.1 |
(C)
|
feed ingredients |
HT |
|
fly maggot protein |
30 |
|
soybean meal (processed GMO soybeans) |
23 |
|
peanut bran |
15 |
|
yeast |
5 |
|
flour |
20.3 |
|
soy lecithin |
1 |
|
fish oil |
1 |
|
soybean oil |
1 |
|
choline chloride |
0.6 |
|
potassium dihydrogen phosphate |
1 |
|
multivitamin |
1 |
|
multimineral |
1 |
|
Vc Phosphatidic Acid |
0.1 |
(D)
Table S2. Amino acid composition of each feed group, unit: %
|
Amino acid type |
CK |
FM |
FM |
HT |
|
Asp |
3.79 |
5.30 |
5.69 |
5.50 |
|
Thr |
1.57 |
2.18 |
2.78 |
1.97 |
|
Ser |
1.42 |
5.08 |
4.99 |
5.32 |
|
Glu |
5.79 |
6.79 |
6.59 |
6.35 |
|
Gly |
1.75 |
3.77 |
3.39 |
3.30 |
|
Ala |
2.20 |
1.86 |
1.69 |
1.73 |
|
Cys |
0.32 |
1.79 |
1.77 |
1.76 |
|
Val |
1.85 |
2.69 |
2.66 |
2.32 |
|
Met |
0.51 |
0.91 |
0.87 |
0.97 |
|
Ile |
1.72 |
2.88 |
2.90 |
2.85 |
|
Leu |
3.00 |
3.28 |
3.14 |
3.19 |
|
Tyr |
1.36 |
2.57 |
2.61 |
2.58 |
|
Phe |
1.58 |
2.60 |
2.51 |
2.70 |
|
Lys |
2.70 |
3.87 |
3.79 |
3.76 |
|
His |
1.07 |
1.56 |
1.78 |
1.50 |
|
Arg |
2.24 |
3.49 |
3.76 |
3.56 |
|
Pro |
1.23 |
4.58 |
4.36 |
4.74 |
Note: Tryptophan was not measured.
Table S3. Nutritional composition of groups of feed, unit: %
|
Nutrient composition |
CK |
FM |
FM |
HT |
|
Crude protein |
30.0 |
31.6 |
30.5 |
31.0 |
|
Crude Fiber |
5.0 |
9.9 |
10.3 |
10.6 |
|
Calcium |
4.0 |
3.9 |
3.9 |
4.1 |
|
Phosphorus |
1.6 |
2.1 |
2.0 |
2.0 |
|
Crude Ash |
16.0 |
10.5 |
10.4 |
10.1 |
|
Water |
12.0 |
40.0 |
35.0 |
10.0 |
Round 2
Insufficient answers -- still recommend reject
[Reply] We are very sorry that we did not respond adequately in the first round of responses, and we have provided a new detailed response to the first round of suggestions.
